# An Analysis of Circular Economy Deployment in Developing Nations' Manufacturing Sector: A Systematic State-of-the-Art Review

**Rajeev Rathi** [1,*] , **Dattatraya Balasaheb Sabale** [1] , **Jiju Antony** [2], **Mahender Singh Kaswan** [1] **and Raja Jayaraman** [2,*]

1. School of Mechanical Engineering, Lovely Professional University, Punjab 144411, India
2. Department of Industrial and System Engineering, Khalifa University, Abu Dhabi 127788, United Arab Emirates
*   Correspondence: rathi.415@gmail.com (R.R.); raja.jayaraman@ku.ac.ae (R.J.)

**Abstract:** Globalization has created a competitive environment in the manufacturing sector in terms of the quality, cost and user experience of the product. The product life cycle has shortened, which adds multiple products to production lines. This has led to adding complexity to the input material, cost of operation and waste generation through the manufacturing system. Circular economy (CE) has a big potential to overcome the manufacturing waste and provides a competitive solution. In the present study, a systematic literature review was conducted to analyze the current state of CE in the context of India and other developing countries. The study explored the status of implementation, benefits and possible avenues for future research. The present study provides a helping hand to industry practitioners and front-line managers to understand CE benefits in their operations.

**Keywords:** circular economy; sustainable manufacturing; re-manufacturing; product recycle

## 1. Introduction

As the consumption patterns have changed, most products are not used throughout their life cycle. The customers' needs are changing over time due to applications and technology. The linear pattern of the product life cycle design, manufacture, application and end of life (Figure 1) creates challenges to the sustainability of the environment. Linear consumption pattern creates heavy demand for products and generates a lot of waste in day-to-day life. A lot of pressure has been put on all input resources to meet the need for the linear life cycle of a product. This model is not sustainable in the long run for the environment and businesses.

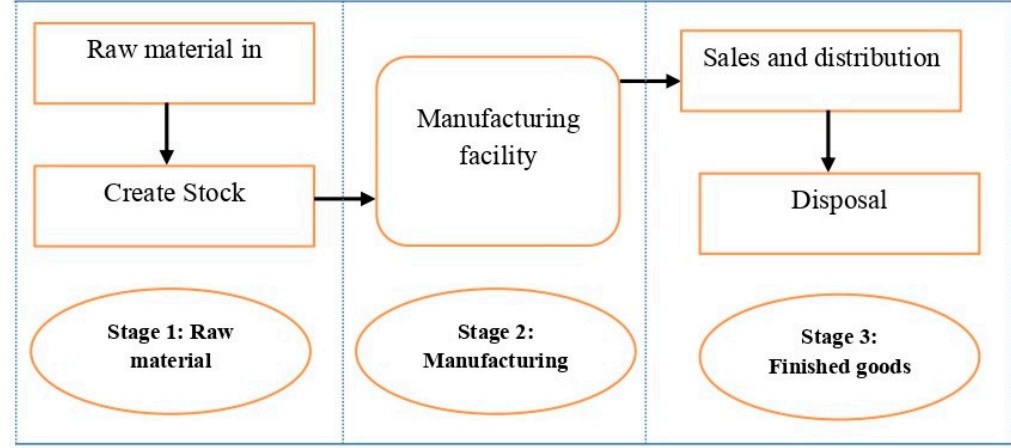

**Figure 1.** Product life cycle.

The environmental impact and limitation of resources have triggered the actions by worldwide bodies to focus on setting up goals toward sustainability. This approach provides benefits for economic, environmental and social engineering. The economic model of utilizing a product or part of it to its fullest application through reuse and recycling in a repeated loop of life is called the circular economy (CE). CE enables organizations to improve resource utilization, abide by government regulations and overcome societal and investor pressure. The CE model provides a leading edge to meet customer expectations and deliver value for money [1]. McKinsey forecast that, globally, CE will save USD 3.7 trillion per year through resource efficiency. Manufacturing systems and material waste generate two-thirds of greenhouse gas emissions. This issue can be minimized through the responsible production CE model. This is a win–win situation on the front cost of sustainability and corporate social responsibility toward the environment.

The CE has been in the focus for the past decade, but it became the center of action in the last couple of years. The MacArthur Foundation is pushing with a green goal intending to focus on sustainability instead of only looking at economic growth. The United Nations and many economic fora have rolled out plans for sustainability. The European Union has strongly put forward the plans to implement CE and expects to generate USD 600 billion per year returns in the European industrial manufacturing sector [2]. Many consultancy reports have shown that big organizations, such as Apple, Samsung, Ford and GE, and CEOs are taking steps toward onboarding CE as an approach to contribute toward sustainability goals [3]. Japan is also forcing the industries to implement the green revolution through the Law of Recycling-Based Society (Ministry of Economy, Trade, and Industry (METI), 2004). The developed nations' policies showcase their commitments toward sustainability, and they have set short-term and long-term goals toward sustainability, whereas the developing nations are lagging on sustainability goals. China has been the leading nation in starting the journey toward circular principles by passing its law on the CE in 2008. India's current economic and population trends showcase that in a couple of decades, it will be the fourth largest economy. This economic growth creates headwinds in terms of the availability of resources, rapid urbanization and high level of poverty. It is very difficult to achieve GDP targets, sustainability and socioeconomic development under a linear economy model. This has come to the attention of many academic researchers, strategists and the industrial sector, but clarity is missing concerning sustainability and economic headroom. Most SME organizations think that sustainability comes with a cost to the company. This derails the journey of many companies toward the CE model. The main objective of this article is to explore the current status of CE adoption in the manufacturing sector of developing nations, especially India, to understand the know-how of the CE, the relationship of the circular economy with sustainability and to identify the opportunities for the Indian industrial sector.

The rest of the article is structured as follows. Section 2 enumerates the adopted methodology to conduct this state-of-the-art literature review. Section 3 presents a systematic literature search methodology and describes the different statistics related to the adoption of CE in the context of developing nations. Section 4 depicts the different CE principles and characteristics. Section 5 illustrates the adoption level of the CE in different organizations. Section 6 presents the current challenges in India and its solutions in the context of CE. Section 7 presents the opportunities for CE in the manufacturing and service sectors of developing nations. Section 8 demonstrates the results, whereas Section 9 presents a discussion. The final section of the manuscript depicts the inferences drawn from the present study.

## 2. Research Methodology

The research methodology in the present research work is presented in Figure 2. Firstly, a systematic literature review (SLR) methodology was defined to conduct a constructive literature study on the different aspects of CE adoption within the context of developing nations, especially India. SLR provides a unique way to survey the literature, so that no

pertinent and relevant article is left out from the literature review [2,3]. A total of more than 200 papers were found on CE from academic scholars and industry researchers along with consultants. Google Scholar, ScienceDirect and ResearchGate search engines were used for searching the articles published on circular manufacturing, with keywords including: circular economy (CE), sustainable methodology, circular manufacturing, re-manufacturing, recycling of product, industry 4.0 and sustainable manufacturing. The articles considered in this study were from the year 2000 to date. To develop the basic know-how and establish the foundation of CE, the authors conducted a descriptive analysis of CE based on the journal, country and theme of each paper. The information on the country of origin of the papers enabled us to comprehend the level of adoption of CE in different nations. Further, to find out how CE adoption can be enhanced, different enablers of CE were found in the literature that propel organizations to introduce CE initiatives for developing the pursuit toward a resilient industry.

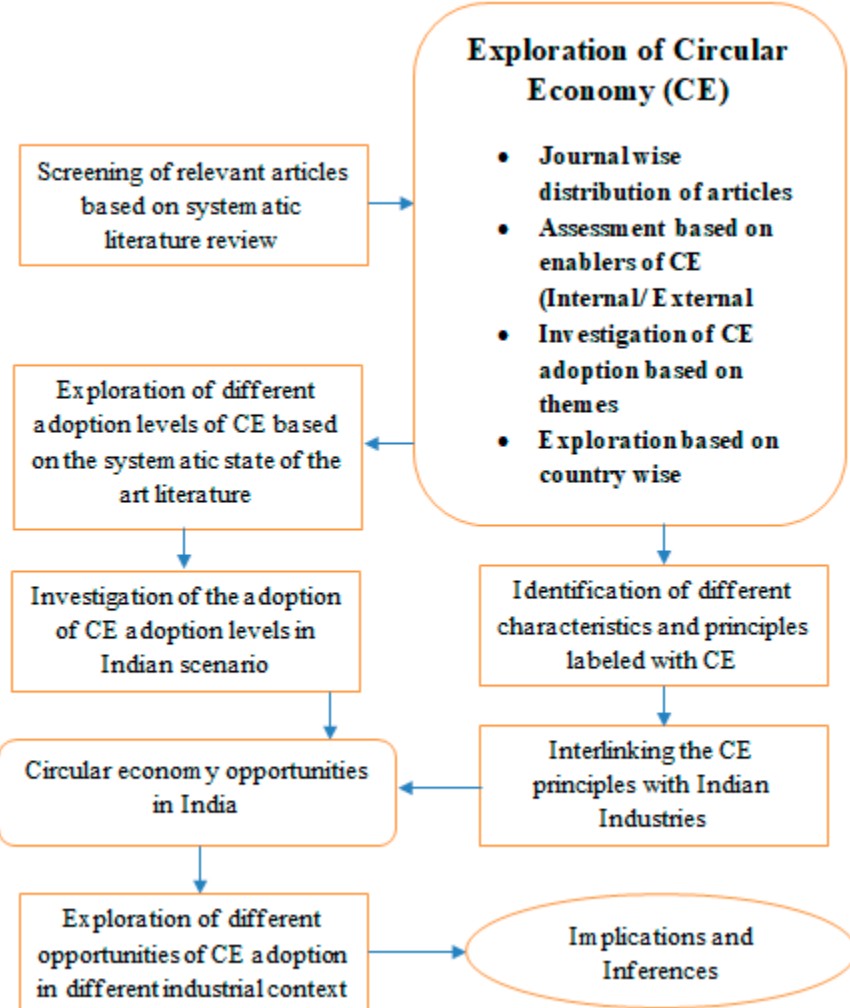

**Figure 2.** Research methodology.

The adoption of new systems, methodology and technology requires supporting factors, which are called enablers. The enablers were listed from different papers published in journals. They were categorized as internal or external based on different criteria. Internal enablers were screened for their relevance to the manufacturing industry and eliminated if not applicable. The hygiene level maintained in the food industry was mentioned as one of the internal enablers, which has no impact on manufacturing and was therefore dropped from the study. External enablers were selected based on the countries where the research was conducted and whether they related to developing nations and were relevant

to Indian consumers. For example, the North American Free Trade Agreement (NAFTA) was mentioned as an external enabler, but it was not relevant to India and thus excluded from the study. The authors also investigated the different themes of circular economies, such as "Circular Economy Drivers and Barriers", the" Circular Economy Framework and Industry 4.0" and the "Circular Strategy and Business Model". The investigation of the articles was conducted to comprehend the level of integration of CE with other business strategies, methods and facilitating mechanisms. This will aid in the exploration of future avenues for research in the context of CE. The systematic exploration of CE articles further aids in the understanding of the principles of the 9 Rs coupled with CE and the different levels of adoption of CE. This further helps in understanding how these principles and different adoption levels can be undertaken by the Indian industries. Further, based on a critical investigation of different articles, the facilitators, principles, themes and different avenues for possible research were explored, which creates a basis for the industry to begin their march toward carbon neutrality. Finally, based on the state-of-the-art literature study, the authors identified different implications for policymakers and practitioners and derived inferences from the illustrated after effects of the study.

### 3. Circular Economy: Systematic Literature Review

A systematic literature review (SLR) was conducted to analyze the status of the CE in the industrial sector. A total of more than 200 papers were found on the CE from academic scholars and industry researchers along with consultants.

The research articles were sorted by using the decision analysis flowchart mentioned in Figure 3. Papers were scrutinized for what fell into the research domain of the CE sphere from the manufacturing sector point of view. Further, the selected papers were filtered by pre- and post-year-2000 era. Papers from before the year 2000 were eliminated from the study, as the CE domain had changed a lot.

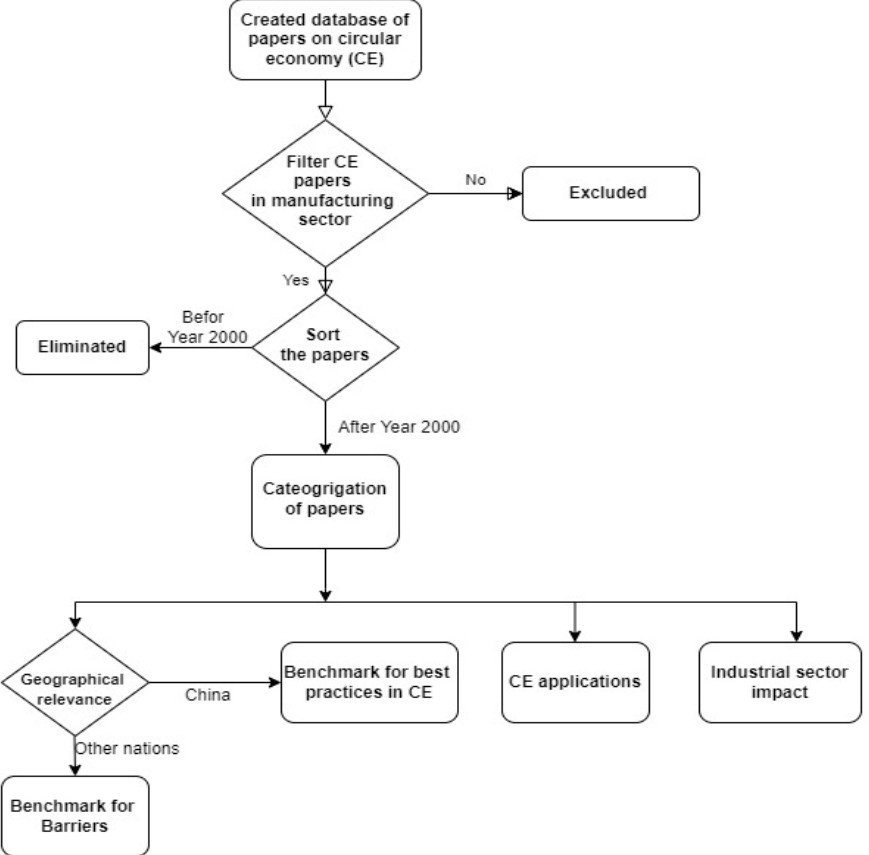

**Figure 3.** Systematic literature review.

A total of 75 papers were selected based on the manufacturing relevance mentioned in Table 1 Further, the selected papers were categorized based on geographical location and CE application domain.

**Table 1.** Circular economy adoption in manufacturing context.

| S.No. | Author | Year | Journal | Research Theme | Research Country |
|---|---|---|---|---|---|
| 1 | Fenna Blomsma. et al. [4] | 2019 | *Journal of Cleaner Production* | Circular Strategy and Business Model | Germany |
| 2 | Federica Acerbi and Marco Taischv [5] | 2020 | *Journal of Cleaner Production* | Circular Strategy and Business Model | Italy |
| 3 | Marco Spaltini. et al. [6] | 2021 | *Procedia CIRP* | Circular Framework and Industry 4.0 | Italy |
| 4 | Heidi Simone Kristensen. et al. [7] | 2020 | *Journal of Cleaner Production* | Circular Economy Drivers and Barriers | Denmark |
| 5 | Biwei Su a. et al. [8] | 2013 | *Journal of Cleaner Production* | Circular Strategy and Business Model | China |
| 6 | A. Buruzs. et al. [9] | 2017 | *International Journal of Environmental and Ecological Engineering* | Circular Strategy and Business Model | Austria |
| 7 | Eva Guldmann and Huulgaard [10] | 2020 | *Journal of Cleaner Production* | Circular Economy Drivers and Barriers | Denmark |
| 8 | Jaeger and Arvind Upadhyaya [11] | 2021 | *Journal of Cleaner Production* | Circular Framework and Industry 4.0 | India |
| 9 | Fernando J. et al. [12] | 2019 | *Ecological Economics* | Circular Strategy and Business Model | Australia |
| 10 | Martin Geissdoerfer. et al. [13] | 2017 | *Journal of Cleaner Production* | Circular Strategy and Business Model | UK |
| 11 | Florian Hofmann. et al. [14] | 2019 | *Journal of Cleaner Production* | Circular Strategy and Business Model | USA |
| 12 | Andrea Urbinati et al. [15] | 2020 | *Journal of Cleaner Production* | Circular Strategy and Business Model | Italy |
| 13 | Adriane Cavalieri. et al. [16] | 2021 | *Sustainability* | Circular Framework and Industry 4.0 | Brazil |
| 14 | Marina P.P. Pieroni. et al. [17] | 2021 | *Journal of Cleaner Production* | Circular Strategy and Business Model | Denmark |
| 15 | Marit Moe Bjørnbet et al. [18] | 2021 | *Journal of Cleaner Production* | Circular Strategy and Business Model | Norway |
| 16 | Kumar, V. et al. [19] | 2019 | *Management Decision* | Circular Economy Drivers and Barriers | India |
| 17 | Bianchini, A. et al. [20] | 2020 | *Sustainability* | Circular Framework and Industry 4.0 | Italy |
| 18 | Giuseppina Piscitelli. et al. [21] | 2020 | *Procedia CIRP* | Circular Framework and Industry 4.0 | Italy |
| 19 | Graziela Darla Araujo Galvão. et al. [22] | 2018 | *Procedia CIRP* | Circular Economy Drivers and Barriers | Brazil |
| 20 | Damien Giurco. et al. [23] | 2014 | *Resources, Conservation & Recycling* | Circular Framework and Industry 4.0 | Australia |
| 21 | Michael Lieder et al. [24] | 2016 | *Journal of Cleaner Production* | Circular Strategy and Business Model | Sweden |
| 22 | Brydges T. [25] | 2018 | *Resources, Conservation & Recycling* | Circular Strategy and Business Model | USA |

**Table 1.** *Cont.*

| S.No. | Author | Year | Journal | Research Theme | Research Country |
|---|---|---|---|---|---|
| **23** | Okoriei. et al. [26] | 2017 | *ESSCP* | Circular Strategy and Business Model | China |
| **24** | Singh. et al. [27] | 2018 | *Resources, Conservation & Recycling* | Circular Strategy and Business Model | India |
| **25** | Graeme Heyes et al. [28] | 2018 | *Journal of Cleaner Production* | Circular Strategy and Business Model | UK |
| **26** | Natalia Marzia Gusmerotti et al. [3] | 2019 | *Journal of Cleaner Production* | Circular Economy Drivers and Barriers | Italy |
| **27** | Ilić et al. [29] | 2016 | *Habitat International* | Circular Economy Drivers and Barriers | Serbia |
| **28** | Rajput and Singh. et al. [30] | 2018 | *Benchmarking: An International Journal* | Circular Framework and Industry 4.0 | India |
| **29** | Himanshu Gupta et al. [31] | 2021 | *Journal of Cleaner Production* | Circular Framework and Industry 4.0 | India |
| **30** | Ana Beatriz Lopes de Sousa Jabbour. et al. [32] | 2018 | *Big Data Analytics in Operations & Supply Management* | Circular Framework and Industry 4.0 | France |
| **31** | Roberto Rocca. et al. [33] | 2020 | *Sustainability* | Circular Framework and Industry 4.0 | UK |
| **32** | Andrea U. et al. [34] | 2021 | *Sustainability* | Circular Strategy and Business Model | Romania |
| **33** | Ana de Jesusa et al. [35] | 2018 | *Ecological Economics* | Circular Economy Drivers and Barriers | Portugal |
| **34** | Valerio Elia. et al. [36] | 2017 | *Journal of Cleaner Production* | Circular Economy Drivers and Barriers | Italy |
| **35** | Brais Suarez-Eiroa et al. [37] | 2019 | *Journal of Cleaner Production* | Circular Strategy and Business Model | Spain |
| **36** | Andrea Cantu et al. [38] | 2021 | *Sustainability* | Circular Economy Drivers and Barriers | Mexico |
| **37** | Andreas Felsberger et al. [39] | 2020 | *Sustainability* | Circular Framework and Industry 4.0 | Austria |
| **38** | Yesim Deniz Ozkan-Ozena et al. [40] | 2020 | *Resources, Conservation & Recycling* | Circular Framework and Industry 4.0 | Turkey |
| **39** | I.S. Jawahir et al. [41] | 2016 | *Procedia CIRP* | Circular Strategy and Business Model | USA |
| **40** | Sunil Luthra et al. [41] | 2022 | *Journal of Business Research* | Circular Strategy and Business Model | India |
| **41** | Marika et al. [41] | 2019 | *Resources, Conservation & Recycling* | Circular Strategy and Business Model | Italy |
| **42** | Shan Zhonga et al. [42] | 2018 | *Resources, Conservation & Recycling* | Circular Framework and Industry 4.0 | China |
| **43** | Julian Kirchherr et al. [43] | 2019 | *Resources, Conservation & Recycling* | Circular Strategy and Business Model | Netherlands |
| **44** | Paolo Rosa. et al. [44] | 2019 | *Journal of Cleaner Production* | Circular Strategy and Business Model | Italy |
| **45** | Bjoern Jaeger and Arvind Upadhyay [11] | 2019 | *Journal of Enterprise Information Management* | Circular Economy Drivers and Barriers | Norway |
| **46** | Enes Ünala. et al. [45] | 2019 | *Resources, Conservation & Recycling* | Circular Strategy and Business Model | UK |

**Table 1.** *Cont*.

| S.No. | Author | Year | Journal | Research Theme | Research Country |
|---|---|---|---|---|---|
| 47 | Azizuddin et al. [46] | 2019 | *Resources, Conservation & Recycling* | Circular Economy Drivers and Barriers | Bangladesh |
| 48 | Bag and Pretorius [47] | 2020 | *International Journal of Organizational Analysis* | Circular Framework and Industry 4.0 | South Africa |
| 49 | Saah and Musvoto [48] | 2020 | *Journal of Contemporary Management* | Circular Strategy and Business Model | Zimbabwe |
| 50 | Martin. et al. [49] | 2021 | *Journal of Cleaner Production* | Circular Strategy and Business Model | Sweden |
| 51 | Korhonen et al. [50] | 2018 | *Ecological Economics* | Circular Economy Drivers and Barriers | Finland |
| 52 | Ünaland Shao [51] | 2019 | *Journal of Cleaner Production* | Circular Strategy and Business Model | China |
| 53 | Testa et al. [52] | 2017 | *Sustainability* | Circular Strategy and Business Model | Italy |
| 54 | Mesa, J.et al. [53] | 2018 | *Journal of Cleaner Production* | Circular Economy Drivers and Barriers | Spain |
| 55 | Dias, V. et al. [54] | 2022 | *Journal of Air Transport Management* | Circular Economy in aerospace manufacturing | Portugal |
| 56 | Kuo, T.C. et al. [55] | 2019 | *Resources, Conservation & Recycling* | Circular Strategy and Business Model | China |
| 57 | Halstenberg et al. [56] | 2017 | *Procedia Manufacturing* | Circular Framework and Industry 4.0 | Germany |
| 58 | Jabbour et al. [57] | 2019 | *Technol. Forecast. Soc. Change* | Circular Framework and Industry 4.0 | France |
| 59 | Jensen and Remmen [58] | 2017 | *Procedia Manufacturing* | Circular Economy Drivers and Barriers | Denmark |
| 60 | Jiliang and Chen [59] | 2013 | *Res. J. Appl. Sci. Eng. Technol.* | Circular Strategy and Business Model | China |
| 61 | Kalmykova et al. [60] | 2018 | *Resources, Conservation and Recycling* | Circular Strategy and Business Model | Sweden |
| 62 | Parida et al. [61] | 2019 | *Journal of Business Research* | Circular Strategy and Business Model | Colombia |
| 63 | Ponte et al. [62] | 2020 | *Journal of Business Research* | Circular Strategy and Business Model | Spain |
| 64 | Pieroni et al. [63] | 2019 | *Sustainability* | Circular Strategy and Business Model | Brazil |
| 65 | Takata et al. [64] | 2019 | *CIRP Annals* | Circular Strategy and Business Model | Japan |
| 66 | Thayla et al. [65] | 2018 | *Journal of Cleaner Production* | Circular Strategy and Business Model | Brazil |
| 67 | Sassanelli et al. [66] | 2019 | *Journal of Cleaner Production* | Circular Economy Drivers and Barriers | Italy |
| 68 | Sauerwein et al. [67] | 2019 | *Journal of Cleaner Production* | Circular Framework and Industry 4.0 | Netherlands |
| 69 | Sarc et al. [68] | 2019 | *Waste Management* | Circular Framework and Industry 4.0 | Austria |

**Table 1.** *Cont*.

| S.No. | Author | Year | Journal | Research Theme | Research Country |
|---|---|---|---|---|---|
| **70** | Tolio et al. [69] | 2017 | *CIRP Annals* | Circular Strategy and Business Model | Italy |
| **71** | Tukker [70] | 2015 | *Journal of Cleaner Production* | Circular Strategy and Business Model | Netherlands |
| **72** | Vimal et al. [71] | 2019 | *Journal of Manufacturing Technology Management* | Circular Framework and Industry 4.0 | India |
| **73** | Zhong and Pearce [72] | 2018 | *Resources, Conservation and Recycling* | Circular Framework and Industry 4.0 | China |
| **74** | Zhou et al. [73] | 2017 | *Journal of Cleaner Production* | Circular Strategy and Business Model | China |
| **75** | Wang, S. and Zhang, Y. [74] | 2018 | *Journal of Advanced Oxidation Technologies* | Circular Strategy and Business Model | China |

The categorized articles show that China is leading in CE-related research and implementation in the manufacturing domain after the government's CE policy introduction in 2010. Papers from China are considered as best practice, as this is a developing nation like India. These data will be used for further analysis of what are the best practices of the CE in manufacturing in leading countries and the shortcomings in the rest of the countries to define the roadmap.

Enablers create a healthy environment for the adoption of any new concept by the organization. Internal enablers were screened for their relevance to the manufacturing industry and excluded if not applicable (Table 2). The hygiene level maintained in the food industry was mentioned as one of the internal enablers, which has no impact on manufacturing and was therefore excluded from the study. External enablers were selected based on the countries where the research was conducted and whether they related to developing nations and were relevant to Indian consumers (Table 3). For example, the North American Free Trade Agreement (NAFTA) was mentioned as an external enabler, but it was not relevant to India and was thus excluded from the study. They were rated as high, medium and low impact based on the scholars' opinions mentioned in the journals. The enablers' rating was decided based on the impact mentioned in the research articles. A high rating was assigned if more than 60% of the articles mentioned it as a high-impact enabler. A medium impact was assigned to an enabler if it was mentioned in less than 60% to 30% of the articles. If an enabler was mentioned in less than 30% of the articles, it was assigned a low impact. Twenty-one internal and eighteen external enablers were found relevant to developing nations, such as India. Hight-impact enablers drive the actions for CE implementation. The results below are preliminary findings from a systematic literature review and recommendations mentioned by the manufacturing SMEs and academicians.

**Table 2.** Internal enabler and its impact on CE implementation.

| Enablers | Impact Scale | | |
|---|---|---|---|
| | Low | Medium | High |
| Type of Business | | ■ | |
| Company Ownership | | ■ | |
| Leadership Vision | | | ■ |
| Organization Goals | | | ■ |
| Optimization Drivers | | | ■ |
| Financial Drivers | | | ■ |
| Product | | | ■ |
| Manufacturing Process | ■ | | |
| Quality Policy | | ■ | |
| Technology Availability | ■ | | |
| Probability/Market share/benefit | | ■ | |
| Stability | | | ■ |
| Customer relationship | | | ■ |
| Innovation roadmaps | ■ | | |
| Brand Value | | | |
| Human Resource and Workforce Mindset | | | ■ |
| Organizational Structure | | | ■ |
| Business Growth | | ■ | |
| Training and Coaching | | ■ | |
| Lean Culture | | ■ | |
| Capital | | | ■ |
| Agility of organization | | | ■ |

**Table 3.** External enabler and its impact on CE implementation.

| Enablers | Impact Scale | | |
|---|---|---|---|
| | Low | Medium | High |
| Government Policy, Rules and Regulations | | | ■ |
| Supply Chain Drivers | | ■ | |
| Social Changes | | | ■ |
| Stakeholder Pressure | | | ■ |
| Infrastructure | | ■ | |
| Globalization | | ■ | |
| Trade Union | ■ | | |
| Professional Association | ■ | | |
| Political Changes | ■ | | |
| Competition | | ■ | |
| Environmental Drivers | | | ■ |
| Per Capita Income | | ■ | |
| Economy and Market Condition | | ■ | |
| Geographical Locations | | | ■ |
| Industry 4.0 trends | ■ | | |
| Voice of Future Customer | | | ■ |
| Resource Scarcity | | | ■ |
| Commodity Pricing and its fluctuation | | | ■ |

The data collected from the research papers indicate that the CE concept started evolving around 2013 (Figure 4). After 2018, the trends indicate that the focus of people and organizations increased toward sustainability goals and finding solutions to the CE concept.

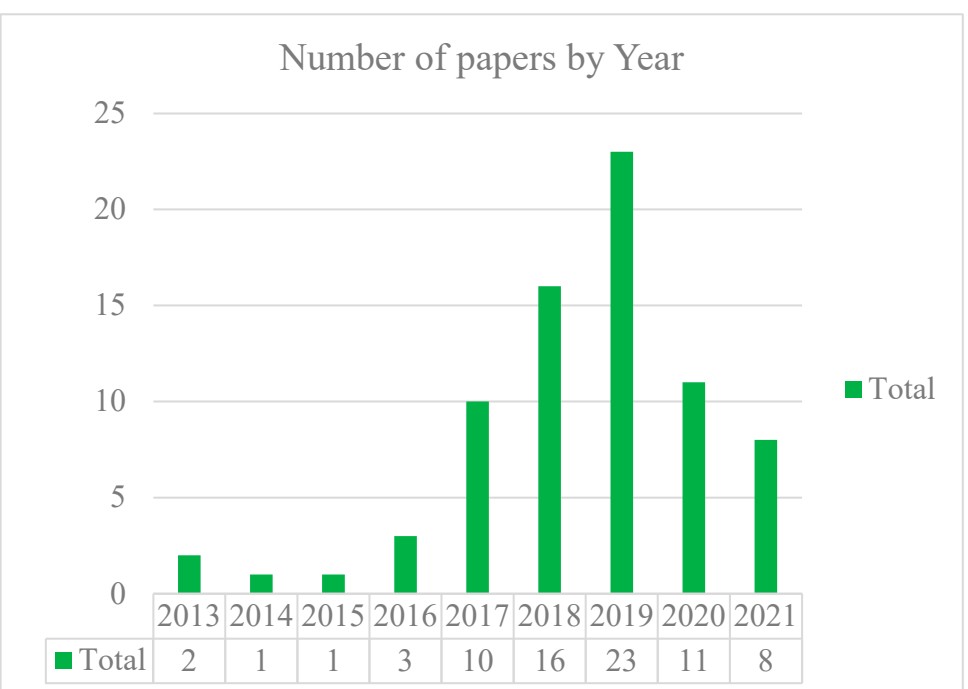

**Figure 4.** Annual paper distribution.

The research papers were narrowed down to the last five years, and 50 papers were shortlisted from reputed journals/publishers, such as the *Journal of Cleaner Production*, *Sustainability*, *Resources, Conservation and Recycling*, *The CIRP Journal of Manufacturing Science and Technology* (CIRP-JMST), *Procedia Manufacturing* (Figure 5). Research trends after 2019 indicate that many industries and institutes have actively worked on the CE concept and experimented with business cases to institute full implementation.

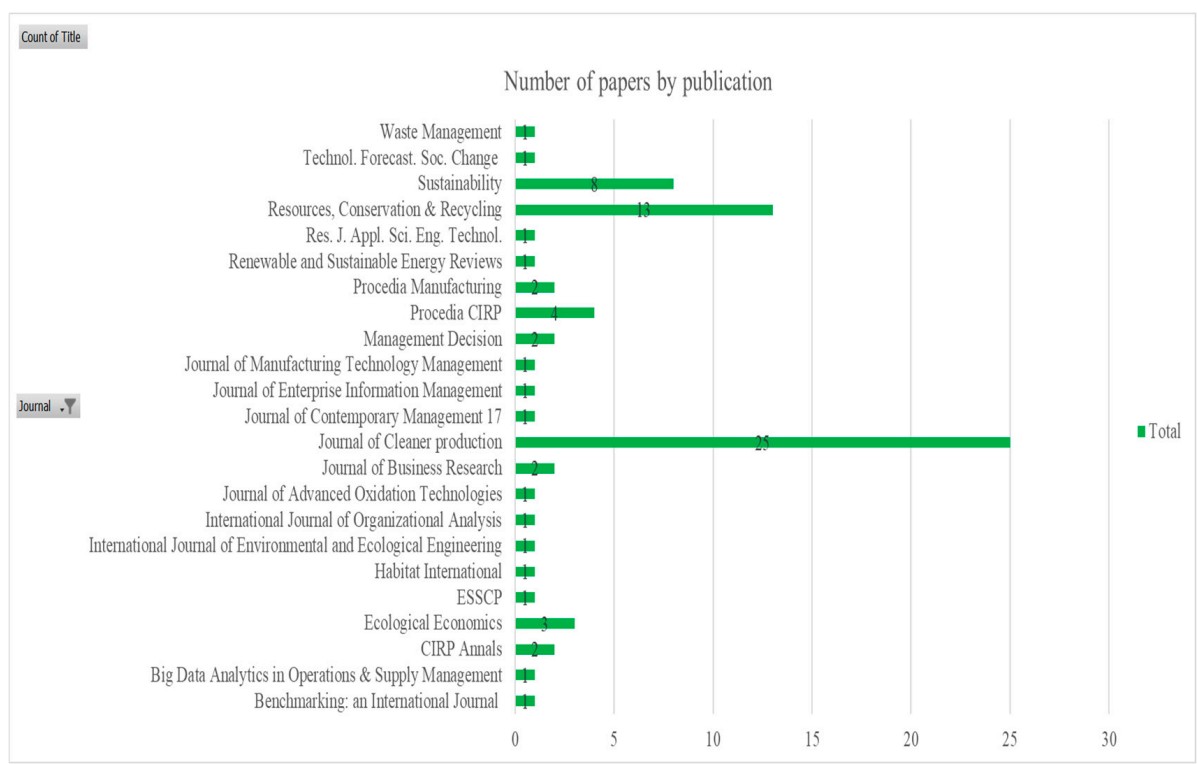

**Figure 5.** Journal-wise distribution of CE papers.

As per the literature, the majority of research works published on the CE emphasized the basic concepts, i.e., the business models, impact on industry 4.0, frameworks, barriers and drivers of the CE (Figure 6). The literature review on the CE provided in-depth knowledge of the concept and definition, the architecture of different models, its application in different domains and the future scope of research.

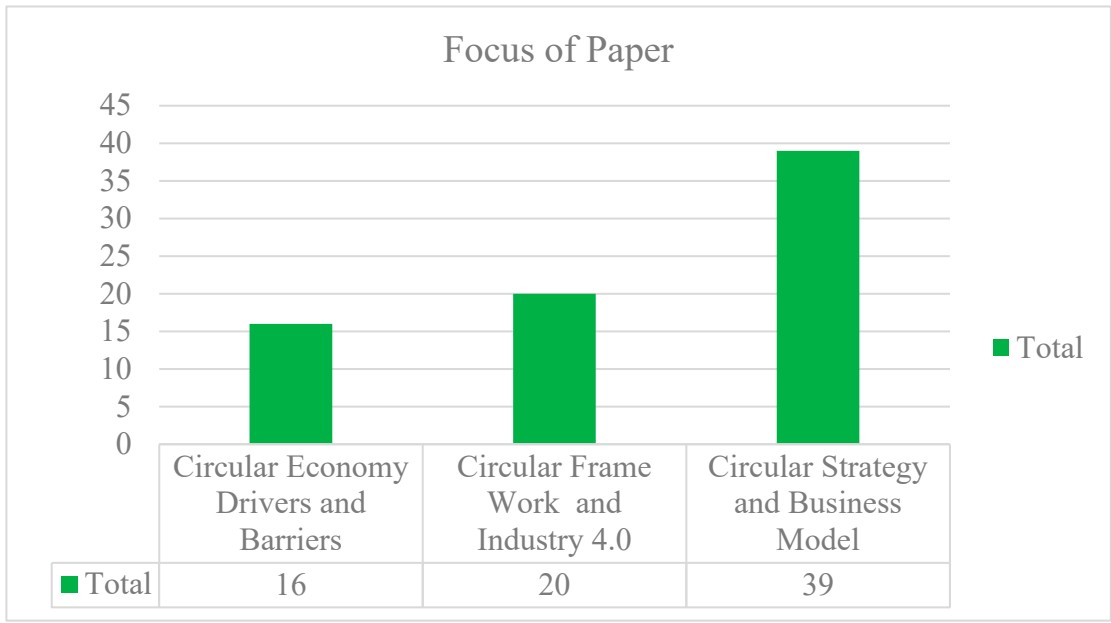

**Figure 6.** Focus area of CE publications.

## 4. Circular Economy Principles and Characteristics

The CE's basic principle is to optimize the utilization of resources or assets throughout the product life cycle. The CE concept encompasses eliminating waste from underutilized life or applicability of a product. CE has replaced the end-of-life phase in the linear life cycle of a product as the core concept of recycling. As this started evolving, new principles came into the picture. In 2004, the Japanese government introduced the "3R Policy", which talks about reducing, reusing and recycling. Later in 2008, the European Union proposed waste hierarchy guidelines for CE practices, with a focus on 4R—reduce, reuse, recycle and recover [15]. As more researchers and strategists started exploring and experimenting with CE practices, different R frameworks have been proposed. Currently, the 9R (Figure 7) principles are used for the implementation of CE.

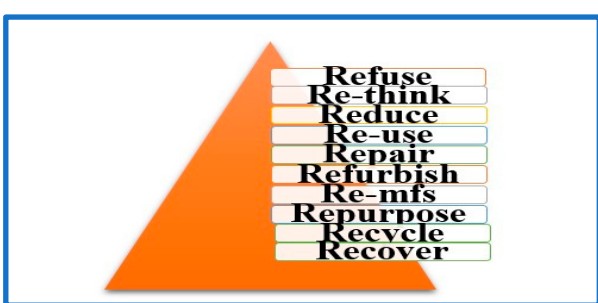

**Figure 7.** 9R framework.

The shift from a linear product cycle to a circular life cycle of products or resources represents a substantial mindset shift. It is very important to understand the guiding principles to make a successful journey. The details of 9R are given below.

*Refuse:* This principle is based on eliminating the need for a product for operation or function [44]. Alternative ways or substitutes are identified to meet the need. An example is petrol used as fuel, which can be refused for automotive purposes with the help of battery-operated automobiles. India will benefit by refusing to use polluting substances and making the environment greener [75].

*Rethink:* This approach involves changing the thinking about the mindset of the application of a product as single use. The product can be designed as a shared product or for multiple needs or applications. An example of this is the adoption of CNC machines instead of special-purpose machines, which are designed for specific needs. The Indian government needs to encourage the consumers and manufacturers to move from the linear to the circular model and think sustainably [76].

*Reduce:* Increasing the efficiency of a product or manufacturing process, so that it will reduce the consumption of natural resources or input to the system. An example of this principle is improving the fuel efficiency of an automotive product to achieve more with less input. Manufacturing efficiency is the ratio of output to input. If we are able to maintain the same input and generate more output, then we will be able to derive maximum value. Indian manufacturers can reduce their dependency on natural resources, input materials and imports by applying this principle.

*Repair:* This principle is based on the right to repair. The product should be designed in such a way, so as to allow the customer to fix the issue in the product and process and extend the life of the product instead of ending it abruptly [77]. Example: Repairing the punctured tire by fixing or replacing the faulty components and continuing its usage for the rest of its life. The right to repair provides the opportunity for Indian consumers to optimize the running cost of the product.

*Refurbish:* This involves restoring the condition of a product or process to its original condition or similar, so that it can be supplied again in the user market for utilization. Example: Old cellphones are collected through an exchange program and refurbished for resale in the consumer market [78]. The refurbished product serves the need of low-income consumers in developing countries, such as India.

*Remanufacture:* This principle is based on making use of a working part of a faulty or life-ended main product in a new or similar product. Example: Using the battery of an old or damaged cellphone for a new or different cellphone. This is going to help the micro, small and medium enterprises (MSME) add new service offerings and grow their business.

*Repurpose:* If a product is designed for one application but is not useful, then utilize it for another application where it fits suitably. Example: Using the working engine of an old automobile as a power generation unit. Indian consumers will be able to utilize one product for multiple purposes with this approach and save on the cost of living.

*Recycle:* Recycling is the core of the CE, where we change the form of a product or its core material and use it as a raw material for another product, which is of low quality but meets the need. Example: Rubber and plastic broken parts are recycled and used as raw materials for a new product. Recycling will aid India in achieving the United Nations' sustainable development goal related to water, land and air.

*Recover:* This principle is based on utilizing a material of no use to recover it in the form of energy [15]. Example: Ethanol is the best example, where the compost of a material is used for the creation of fuel. This principle will help Indian manufacturers convert non-useful products and recover them in the form of raw materials, reducing the input cost.

All these principles are the basis for the implementation of the CE at any level of implementation. The formulation of a CE roadmap along these guiding principles will help in the success of the program.

## 5. Circular Economy Adoption Level

The adoption of CE is achieved at various levels based on the scope of implementation. There are three basic levels (Figure 8): product-based approach, organization-based approach and regional- or national-compliance-based approach. The product-based approach

is generally based on designing products for refurbishment, remanufacturing and repair instead of ending the product life [79,80]. This approach provides the optimal use for products with acceptable quality. The organizational-level approach is derived through economic value and corporate social responsibility. Most organizations have set goals for sustainability through which they emphasize recycling natural resources in their daily operations. Organizations also focus on the environment in their design through product life cycle management. The third level is macro-level adoption at the national or regional level where the commitment toward the environment drives the changes. Worldwide fora, such as the United Nations, have adopted the Sustainable Development Goal (SDG12) of sustainable consumption and production, whereas the European Union promotes the 4R framework—reduce, reuse, recycle and recover. Seventy-nine countries and the European Union have one national policy to promote SDG12 [14]. These national policies and regulations are one of the driving factors for the circular economy. Florian Hofmann. et al. shared the importance of the "Think global—Act local" strategy for circular business models (CBM) for organizations that operate worldwide and serve different consumer bases. Paolo Rosa. et al. provided a comparative study between the MacArthur Foundation's Resolve (regenerate, share, optimize, loop, virtualize and exchange) framework and other CBM models to decide on the adoption levels.

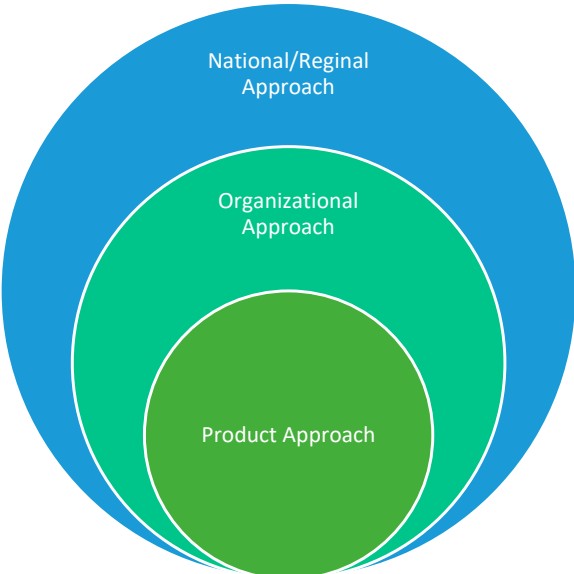

**Figure 8.** Circular economy adoption level.

The developing countries need to start with the national approach by bringing in laws of circularity. The government needs to provide subsidies and incentives to manufacturers and consumers [81]. Recently, the Indian government has started giving tax benefits to customers purchasing electric vehicles. This example shows how the national and regional approaches help in creating an ecosystem for the CE. Many Indian startups are using the leased or pay-per-application business models, which offers a rethought approach to usage. Orix, in collaboration with Hyundai, offers a car lease option instead of purchase. This promotes circularity, as after the lease period, the customer gives the vehicle back to the manufacturer. This Orix model is an organizational approach to circularity. Creating an alternative to petrol fuel is a product-based approach. India is substantially dependent on other countries in terms of the demand versus its natural fuel availability. The Indian Ministry of Petroleum and Natural Gas (MoPNG) revealed the use of ethanol in petrol to reduce this dependency and reuse the waste from the sugar industry. These different adoption levels and initiatives are small milestones toward the long-term success of the developing countries.

### 6. Circular Economy Opportunity in India

India's GDP is consistently showing upward trends and will be in the top five economies. India is one of the largest consumer markets by population, and the industrialization is growing very rapidly. Material consumption after the year 2004 showed a rapid increase and will be on an upward trend based on the GDP forecast. The current scenario demonstrates the strain on resources, which creates a big question concerning the availability of resources in the future. The forecast shows that the requirement for raw materials will be approximately 15 billion tonnes by 2030 and close to 25 billion tons by 2050 [82]. These figures indicate that input resources demand on metal, energy and mineral will grow three-fold according to the medium growth scenario. In the current scenario, India is heavily dependent on the import of critical raw materials, and this is going to grow, as India suffers from limited resources, unavailability of technological advances and issues relating to mining conflicts [83]. As the CE is the only logical long-term solution to match the supply and demand of the Indian growth forecast, creating awareness among the consumers of the 9R principles is key to its adoption. Indian manufacturers and products sold in India need to be compatible with the 9R principles, which will enable circularity. This is the right time to redefine the policies for future growth concerning the consumption patterns and local socioeconomic conditions in India. Policies should be formalized, keeping in mind sustainability, minimal impact on the environment and benefits to all income levels of the population. Material consumption is shown in Figure 9.

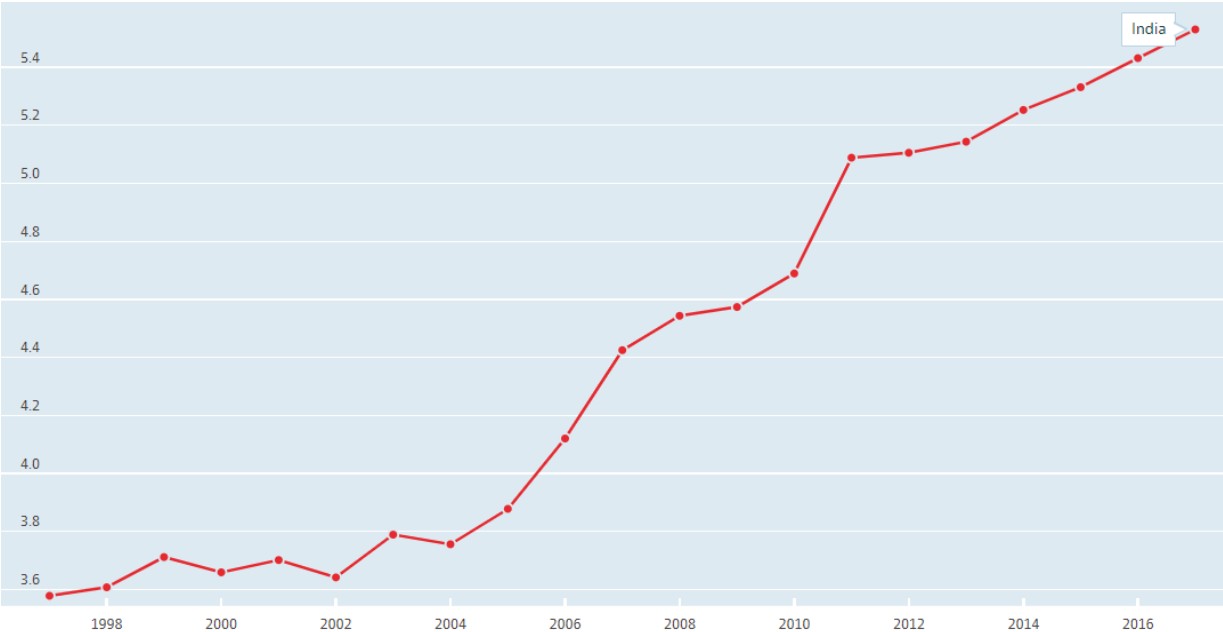

**Figure 9.** Total material consumption: tonnes/capita, 1997–2017.

NITI Aayog is responsible for long-term planning in India and is a strong advocate of CE adoption. They have recommended the government of India form policies for CE implementation [84]. The government has established an Indian resource panel to study the trends in resource requirements and issues in the availability and utilization of resources. A study of CE models adopted in Europe will help identify the enablers and implement them as best practices in the Indian journey context. The barriers and lessons learned from emerging markets will help India in forming better CE policies. Both the external and internal factors in the different consumer markets will guide India to plan a stepwise guide for CE adoption in the country.

## 7. Circular Economy in Different Industrial Sectors

The manufacturing sector is constantly contributing 14 to 17% of the world's GDP according to the World Bank [24]. The manufacturing sector plays a vital role in the socioeconomical betterment of the world. Manufacturing companies are involved in the life cycle of a product, from birth to end of life (Figure 10) [85]. This sector generates a lot of waste through its day-to-day processes, which is mostly linear [86,87]. The cost of operation and input material cost in manufacturing have a bigger stake in the success of an organization. Production processes and material consumption in the manufacturing sector are the major contributors to sustainable development. The United Nations' sustainable development goals for 2030 are relevant to the manufacturing sector and important for achieving sustainable development. The manufacturing sector must make a strategic shift from the linear consumption pattern to deal with the increased material cost and unavailability of resources to become successful in a competitive environment. To deal with the challenging era of resources, the CE model will help the manufacturing industry and has high potential benefits for sustainability.

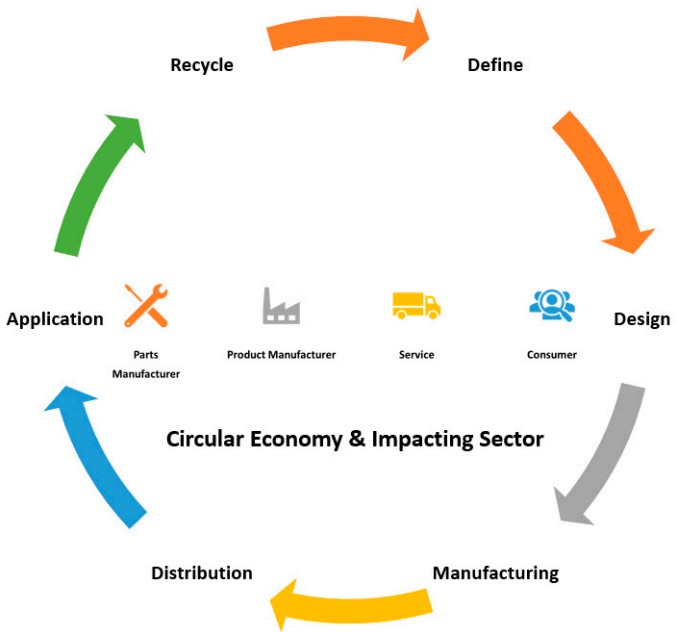

**Figure 10.** Circular product life cycle and impacted sectors.

Many consultancy reports have shown that big organizations, such as Apple, Samsung, Ford and GE, and CEOs are taking steps toward onboarding CE as an approach to contribute toward sustainability goals [3], whereas for the small and medium-sized sector, the benefits of the CE model and its implementation are unclear. The CE is an umbrella term that offers a different strategy for application in different fields. Circular manufacturing is one of strategies under the CE, which improves the ecosystem of the manufacturing sector. Federica Acerbi et al. shared their research concerning the manufacturing area and how the 3R (reuse, remanufacture and recycle) principles are applicable in manufacturing industries [2]. A. Buruzs. et al. mentioned that awareness among consumers of recycled and refurbished products has increased and stimulated more demand, which requires the manufacturers to design products from that perspective [6]. Yesim Deniz Ozkan-Ozena. et al. talk about the many CE opportunities available in the supply chain and distribution of raw materials and finished goods [40]. This concept has untapped potential for the manufacturing sector to become competitive, customer centric and sustainable in the Industry 4.0 era.

The service sector plays a significant role in the product life cycle. The service sector includes the dealer, distribution network, customer support and IT network and involves

making the consumer experience better. It connects the consumer or end user with the manufacturer. They can help the implementation of the CE by servicing the customer and advocating for sustainable development. The service sector influences the application and end-of-life phase of a product. According to the World Bank, the service sector is the highest contributor to GDP; its contribution in the last 10 years has always been above 60% of total GDP (Figure 11).

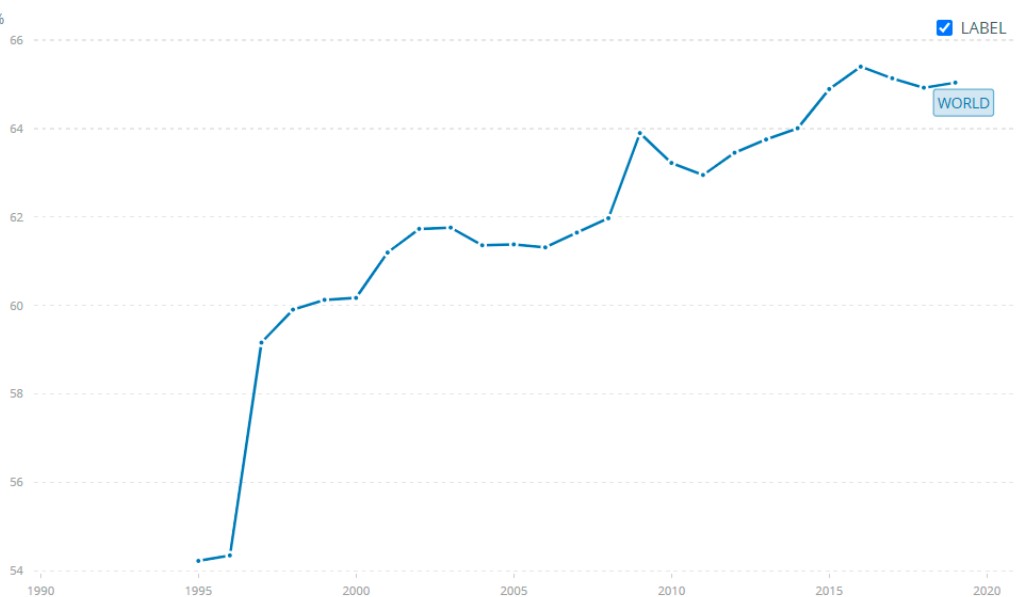

**Figure 11.** Worldwide service sector GDP contribution (Source: World Bank).

The service sector is an exceptionally large entity. In India, trade and repair contribute to 12% of the total service sector (Figure 12). Apart from this, the information technology services also contribute to the betterment of society through the application of technology. Interconnected products with the help of the internet and technology are growing year by year. Smart products are creating a huge demand for electronic and computer-related hardware along with availability. They produce a lot of E-waste in the environment. E-waste has a largest share in the total waste of the world and grows by 35% on a yearly basis [28]. The circular economy model will help the service sector overcome the challenges of imbalance in demand for electronic components, their material and E-waste. Graeme Heyes a. et al. shared their thoughts that technology services and their manufacturers have a high potential to contribute toward circularity. The authors proposed a Backcasting and Eco-design for CE (BECE) framework for computer and technology services [28]. Pieron et al. threw light on the circularity models in the furniture manufacturing industry and services that promotes pay-per-use and leasing models, which drive the circularity of furniture products [17]. The author Arnold Tukker shared his research on the growing middle class and how the product service systems (PSS) will lead to responsible consumption and circularity [71]. In the current state, it is very unclear which CE model can be leveraged for the implantation of the CE, as the service sector is very scattered. Additionally, one model will not apply to all the segments of the service sector.

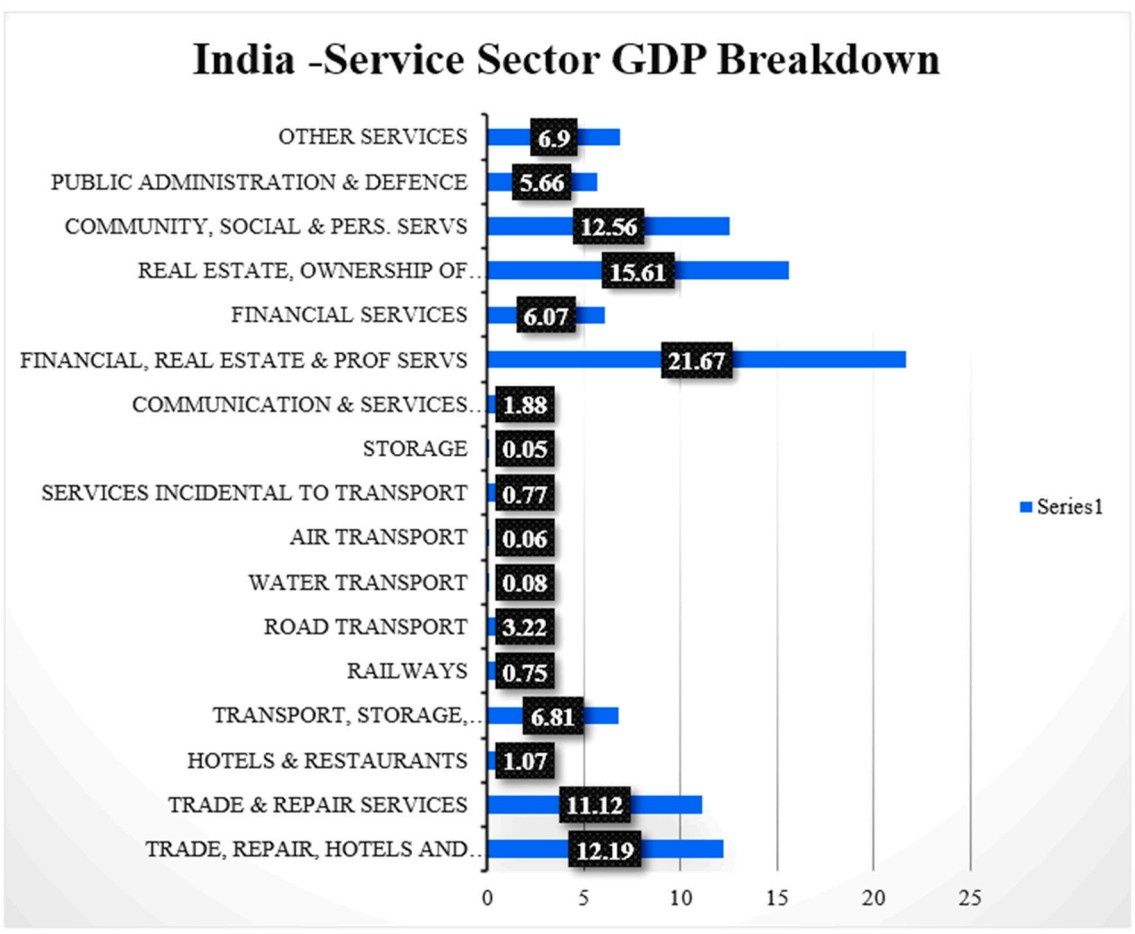

**Figure 12.** Indian service sector GDP breakdown (Source: NITI Aayog, India).

### 8. Results

The CE and the associated aspects of green technology have brought significant changes in the management of wastes, emissions and precious organizational resources [76,88]. It was found that the adoption level of CE aspects is still in its infancy in the developing nations. It was identified through a comprehensive review of academic literature that there is still hesitancy regarding the CE model adoption in manufacturing in the developing nations due to missing clarity on what, when and how to adopt from the ocean of CE models. The focus on the design of a CE is missing in the research articles. All researchers mentioned the 3R to 9R guiding principles for the adoption of CE but nothing about the birth of the products. The success of CE product life cycle depends on designing products for circularity. Further, one CE model does not apply throughout the world in an entire sector. A CE model in agriculture or retail does not suit manufacturing and the related service sector. There is a lack of information on the architecture and roadmap for CE implementation in manufacturing. Moreover, CE implementation has gained focus in Fortune 100 companies, but the MSMEs in India are still behind due to limited accessibility to research, information and implementation guidance in the CE domain. Moreover, in India, there is a lack of awareness of the benefits of CE in manufacturing. Very few articles have been published specifically in the Indian manufacturing sector context. With information based on the Indian consumer patterns missing, the model for the CE leads to derailing CE efforts in the Indian manufacturing sector.

The Indian industry is forecast to grow rapidly, with an exponential increase in input material. This balance of growth and sustainability is difficult under the linear model. The Indian industry must adopt circular business models. As distribution has the biggest share in the industrial operation, as it involves raw materials to finished goods, designing the

green supply chain network coupled with CE will help industries with big cost savings and meeting environmental goals. Further, CE recovery and recycling facets will bring the unused components, material and waste to another value chain. Indian industries will benefit in terms of the reduction in waste and optimal use of resources. Moreover, the longer life of the products will circulate the products from one economic layer of society to another layer. This provides an opportunity for the Indian industry to gain a market share in untapped consumers and serve the need of all consumers instead of producing new products. As most products that are needed occasionally are ideal for most of their life span, the utilization of such items will be maximized if we share rather than own them. The Indian industry has a lot of scope to use the B2B sharing model and thus invest less in assets.

Starting the journey toward the CE is not an easy task, as barriers hinder the path. An organization's efforts in adopting a new methodology or technology come with numerous challenges from the environment where it operates. The major barriers to CE adoption mentioned by different scholars are financial barriers, where originations struggle to justify the return on investments, societal barriers related to consumer patterns and behavior, lack of knowledge and technology availability, government policy, which mainly lacks support, taxation and incentives [22] and institutional barriers, including the infrastructure within an organization for recycling, packaging and waste processing related to product circularity [10]. Every business model, day-to-day operations and customer–partner ecosystems are different, so challenges or barriers that come across may be unique to the general CE barriers found in the literature review [89]. The why–why analysis technique and system failure mode analysis can help the organization examine the barriers in their CE journey.

## 9. Discussion

The adoption of CE practices has not tapped its full potential in the manufacturing sector of developing nations, such as India. To comprehend and foster the strengthening of adoption, the authors conducted a systematic literature review and unearthed different facets that prompt CE adoption in the manufacturing sector. A systematic investigation of the different enablers that encourage the industry to adopt the CE was carried out. Enablers create the ecosystem inside an organization for the implementation of any new concept. The driving factors are very important for a successful adoption of new practices. This assists in gaining a consensus with all the stakeholders and delivers value to the organization. Enablers bridge the gaps in execution and overcome the barriers to adoption. The literature review was conducted to highlight the two levels of enablers—internal and external [6]. The internal ecosystem is an integral part of the organization, which develops the culture and support to enable the adoption of the CE. External drivers are sometimes legitimate actions needed by an organization to abide by the rules and regulations, which foresee the enablement of CE in the organization. Twenty-two enablers were listed under internal, and eighteen enablers are listed under external. Leadership vision, financial goals, product stability, organization structure and workforce mindset are found to be high-impact drivers within the organization [82]. External enablers are also important. Government policies, rules and regulations are one of the enablers for companies to start adopting the CE model to operate in a particular country. Otherwise, organizations will be panelized [22]. Along with this, the social, environmental and stakeholder pressure and rising commodity prices also play an important role and act as an enabler to moving organizations toward CE adoption [35]. The validation of all these drivers is particularly important.

A risk-taking attitude, innovative thinking, CE knowledge, 9R principles and a supporting ecosystem help the organization in making sound decisions regarding the barriers and leverage enablers toward adopting the CE model [77]. Practicing refusal to non-compliance with the environment in daily shop floor operations will help the Indian industries reduce pollution in the air, water and soil. Rethinking, remanufacture and recycling approaches for designing products for circularity is a great step toward changing products from birth and making them more acceptable in the global environment [90]. This enables Indian-manufactured products to be acceptable in the European market, where the

consumer is very cautious about circularity. Reduction is the key to the success of circular manufacturing practices, which encourage the reduction in usage of natural resources and waste from manufacturing. This helps Indian manufacturers reduce the input costs and limit the natural resource dependency. Reuse, recovery and repurposing enables parts from one product to be used for another application, where the requirements are different than the original application. This helps the manufacturer reduce the product development time and achieve speed to the market. Refurbishment serves the need of different economical layers in society, which is very important for developing countries, such as India, where per capita income is low. Refurbished products are reconfigured to their original state and make a re-entry into the consumer market. Integration also plays a significant role in easing out the hurdles in implementation. Both vertical and horizontal integration at the multi-organizational level with supply chain partners, customers and employees at different hierarchy levels has a higher impact on the success of circular economy implementation.

A comprehensive study of the principles and characteristics of CE leads to an increased know-how of the industrial practitioners and managers that will further facilitate the organizations to implement CE measures in their business practices to foster environmental sustainability. The present research work provides the know-how on the 9′R principles of CE that can be utilized by industrial organizations. Further, the study also looks into different models and frameworks of CE and provides a detailed investigation of the same. This will further help practitioners decide which kind of model can be best suited for a particular organization. Moreover, the opportunities for the service and manufacturing sectors were also explored. This will further add to exploring which areas need to be undertaken within the scope of CE adoption. The present thorough detailed investigation also explored the different levels of CE adoption, thus providing guidelines to the industrial managers on what kind of measures must be taken at different levels of CE adoption to make the industry more resilient and sustainable.

## 10. Conclusions and Future Scope

India has a tremendous opportunity to add value through CE adoption. The systematic review indicates that by 2030, India will become a USD 10 trillion economy, half of which could be achieved through CE implementation. The present study presents a systematic state-of-the-art literature review to investigate the measures and levels of CE implementation in Indian industries. The study critically investigates different adoption models of CE in different nations and also provides a systematic know-how of CE. The utilization of natural resources following the 9R principles will ensure optimization and reduce the imbalance between the growing needs of the population and the availability of resources. This CE model will serve the varying needs of economic layers in the population of the world and especially in developing countries, such as India. The Industrial 4.0 revolution started along with CE adoption in bigger organizations with the help of consulting firms. MSMEs are limited due to the cost of leveraging consultancy firms. MSME needs are different to multi-billionaire and multinational operating organizations. MSMEs struggle with the know-how on the CE, as there is no such roadmap and architecture available to leverage in starting the CE journey. The major implication of the present study lies in suggesting a direction for industrial managers, practitioners and policymakers to comprehend the circular economy level of adoption, potential challenges and different investigation measures of the CE model in the context of developing nations, especially India. The study will enable industrial organizations to curb their current level of carbon emission through the step-by-step implementation of CE principles in their business functions.

This study also provides future avenues for research. Researchers in the future can investigate the modeling of the circular economy enablers in the product development process of MSMEs. Moreover, the exploration, assessment and mitigation of circular economy barriers can also be undertaken to boost the principles of CE. Further studies can also be conducted to develop the circular economy model with the 9R principles to improve the environmental and other performance metrics of manufacturing enterprises.

Moreover, the study also encourages researchers to develop a comprehensive model of CE for the developing nations to reduce their current emission levels and achieve the net zero carbon target. In the future, studies can also be directed to integrate the CE model with the existing model of Industry 4.0 to make the production system more responsive to prompt mass customization of products. Moreover, researchers can also investigate integrating the aspects of Industry 5.0 and CE through the lens of environmental aspects for improved organizational performance.

**Author Contributions:** Conceptualization, Methodology, Testing, Supervision, Validation, Initial draft writing, review and edit: R.R., Data collection, analysis, methodology, resources, Initial draft writing: D.B.S.; Idea, Project administration, Supervision, Visualization, Review and editing: J.A.; Data collection, analysis, methodology, resources, Initial draft writing and editing: M.S.K.; Project administration, Funding, Validation of results, Visualization, Review and editing: R.J. All authors have read and agreed to the published version of the manuscript.

**Funding:** This publication is funded by the Khalifa University of Science and Technology under Award No. RCII-2019-002–Center for Digital Supply Chain and Operations Management.

**Institutional Review Board Statement:** Not applicable.

**Informed Consent Statement:** Not applicable.

**Data Availability Statement:** The data presented in this study are available on request from the corresponding author.

**Conflicts of Interest:** The authors declare no conflict of interest.

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
