# Peer review of "An Analysis of Circular Economy Deployment in Developing Nations’ Manufacturing Sector: A Systematic State-of-the-Art Review"

_sustainability, doi:10.3390/su141811354_

Round 1
Reviewer 1 Report
Author(s) in the present study attempted to review the circular economy deployement in developing nations' manufacturing sector, the study seems interesting, and in line with the scope of journal, I have few below mentioned suggestions, author(s) may like to work on:
Changes which must be made before resubmission of the revised version
1. Kindly check the reference consistency throughout the article.
2. Discussion section only offers the results. I am not convinced with this. The discussion should offer the significance of the results.
3. Abstract is too large and should be concise and clear.
4. Section 4 and section 6 are missed.
5. Overall proofreading of the article is needed.
Reviewer 2 Report
Dear authors,
thank you for your submission absolutely in line with the last worldwide updates. Although the topic of the article is interesting, some refinement should be done to make it ready to be published.
Introduction:
§ It is suggested to better clarify the research objective of this contribution in the introduction section. Indeed, it is roughly written only at the end of this section without entering in its details.
§ It is suggested to clarify the structure of the manuscript at the end of the introduction section to guide the reader
Methodology
§ At the current state of the manuscript, a methodology section is missing while it has been integrated as part of the literature review chapter. It is suggested to create a separate section where it is possible to clarify the research methodology employed and the research design.
Literature review
§ It is not clear which is the reason why the authors decided to perform a literature review on circular economy as a whole, since many contributions have been already published about it. It might be better to narrow down the focus on developing countries as suggested by the title of the manuscript.
§ It the table it might be suggested to insert the country where the analysis took place in case an empirical study was conducted and the discussion after the table might be focused on the developing countries and their current state.
Chapter 4, 5, 6 and 7:
§ They might be extended based on the findings from the previous literature review. For instance, (Federica Acerbi and Marco Taisch [5]) in the table is focused on manufacturing thus it could be used to elaborate the results in section 5.
Results and Discussion:
§ It is suggested to narrow down the focus on developing countries such as India. The analysis conducted until now seems to be general on circular economy and not specifically on developing countries as the title and the objective suggest.
Conclusions:
§ It is suggested to better clarify what are the contribution to practice and to theory of this research and highlight the future research opportunities based on the findings
General comments:
§ Check the references style since sometimes it is “authors plus date” other times instead references are “numbers”.
Reviewer 3 Report
The paper offers a broad overview of circular economy principles through a systematic literature review, but does not fully translate these to recommendations for the context of study, India. The analysis of the data from the literature review is well presented and relevant/interesting (though I do not think Table 1 is required). The 9R framework is a good summary of the analysis, and could/should be used later in the paper to frame the discussion of the Indian context, in the final section of the paper. Also the discussion of 'two levels of enablers' is introduced on the final page, but should have been introduced also earlier in the paper. This could also be used to frame the discussion about the Indian context.
Round 2
Reviewer 2 Report
Dear authors,
thank you for the revised version of the manuscript. Although some of the comments were addressed, there are still some open points to improve the quality of the article making it publishable. These are reported below.
1) methodology: it is suggested to improve the explanation of the research methodology employed (e.g. analysis dimensions used such as the enablers and the reason behind this choice). Until now, it seems a description of the structure of the study
2) literature review:
-it is suggested to improve table 1 content. In particular, the research country should be the one where the research took place (if any). Moreover, and some errors emerged (for instance, in my humble opinion [33] should be Italy, as well as [34]).
- It is suggested to better explain where the enablers come from (from a previous study? or is it part of the discussion of the results from this study?) and how the analyses of the impacts have been conducted
3) Chapters 4 and 5 should be better linked to developing countries as the rest of the article
Author Response
PFA the response to the comments below.

Round 3
Reviewer 2 Report
Dear authors,
thank you for your revised version of the manuscript. Although most of the issues have been addressed, there are some minor points to be improved.
1) the yellow part in the literature review (results or discussion) are revisions of the previous text allowing to include reasonings and findings about India and other emerging countries. Although the text is there, there are no references that support these sentences and these considerations. Do these findings come from the state of the art of extant literature or are part of your considerations only?
2) it is still missing a discussion section. Maybe part of the considerations above mentioned can be considered a sort of discussion section. it is suggested to revise the structure separating the results from the discussions.
3) regarding the conclusions, it might be improved the definition of the research agenda.
4) there are some typos throughout the text (e.g. line 163 "and" written twice) that should be fixed before publishing
5) the acronyms are used in a confusing way (e.g. CE is sometimes used while in other cases the entire word is written as "circular economy"). I would suggest continuing for instance to use CE once defined for the first time in the text.
Author Response
Dear Reviewer,
Find the responses in the attached file.
